# Towards Optimal Adapter Placement for Efficient Transfer Learning

## Abstract

Parameter-efficient transfer learning (PETL) aims to adapt pre-trained models to new downstream tasks while minimizing the number of fine-tuned parameters. Adapters, a popular approach in PETL, inject additional capacity into existing networks by incorporating low-rank projections, achieving performance comparable to full fine-tuning with significantly fewer parameters. This paper investigates the relationship between the placement of an adapter and its performance. We observe that adapter location within a network significantly impacts its effectiveness, and that the optimal placement is task-dependent. To exploit this observation, we introduce an extended search space of adapter connections, including long-range and recurrent adapters. We demonstrate that even randomly selected adapter placements from this expanded space yield improved results, and that high-performing placements often correlate with high gradient rank. Our findings reveal that a small number of strategically placed adapters can match or exceed the performance of the common baseline of adding adapters in every block, opening a new avenue for research into optimal adapter placement strategies.

## 1 Introduction

Transfer learning is one of the key techniques in modern deep learning, frequently used to reduce the cost of training and to provide better generalization in the low-data regime (Zhai et al., 2019; Liu et al., 2022; Mustafa et al., 2020). In transfer learning, pre-trained neural networks are fine-tuned to solve a new, often related, task. However, fine-tuning the entire model is costly in terms of compute, memory, and storage, especially considering the size of modern deep learning networks (Dehghani et al., 2023; Touvron et al., 2023; Chen et al., 2023b; Chowdhery et al., 2023). This lead to the proliferation of parameter efficient transfer learning (PETL) methods (Pan et al., 2022; Edalati et al., 2022; Hu et al., 2022; Mercea et al., 2024; Pfeiffer et al., 2020b; Hao et al., 2024; Jie & Deng, 2022), that aim to reduce the memory and storage cost by approximating the full fine-tuning solution using significantly less parameters.

One of the most popular approaches in PETL are *adapters* (Houlsby et al., 2019; Rebuffi et al., 2017) – small modules based on low-rank projections which are inserted into pre-trained models to facilitate transfer learning while minimizing parameter overhead. While both the original and new parameters contribute to the output calculation, only the parameters of the adapters are updated during the optimization process. This results in a reduced memory footprint, making it practical to train large networks with limited resources.

In essence, adapters may be interpreted as a way to mimic the fine-tuning of a group of layers using smaller modules. Therefore they are often added *parallel to* or in-between existing layers. At the same time, existing work on transfer learning shows that layers in a pre-trained network have different importance during fine-tuning and that their parameters affect the performance in different amounts (Chatterji et al., 2020). Despite this observation, however, adapters are typically added *uniformly* across the whole network, irrespective of these layer-specific differences. Consequently, we suspect such a practice to be sub-optimal and hypothesise that task specific adapter placement could further improve the fine-tuning performance and efficiency.

Motivated by the above, we investigate the effectiveness of adding adapters at different locations within a neural network. Our work highlights the importance of adapter placement and its influence on transfer performance. Our main contributions are:

- We verify that placing adapters in different layers leads to significant variations in test accuracy, with the effectiveness of each placement differing substantially. Furthermore, the distribution of the optimal adapter locations varies across tasks.

- We expand the search space for adapters beyond standard parallel and sequential placements by introducing long-range and recurrent adapters.

- When placing a single adapter, we observe that the recurrent adapter consistently performs best. The extended search space is also beneficial when adding multiple adapters, with even a small random search leading to improved results compared to the baselines.

- We investigate various metrics for identifying optimal adapter locations within the proposed search space and find that the rank of the gradient correlates most accurately with the final performance. Furthermore, a greedy strategy based on selecting adapters according to such rank not only outperforms alternative selection policies but also requires significantly less computation, paving the way for efficient exploration of optimal adapter placements.

## 2 RELATED WORK

### 2.1 PARAMETER EFFICIENT TRANSFER LEARNING

The goal of Parameter Efficient Transfer Learning is to adapt a pre-trained model to a downstream task while minimizing the number of trainable parameters (Han et al., 2024). A common approach in PETL are *adapters* – small trainable modules inserted into the pre-trained network (Rebuffi et al., 2017; Houlsby et al., 2019; Bapna et al., 2019), which have proven to be effective in various applications, including multi-task learning (Pfeiffer et al., 2020a;b; Stickland & Murray, 2019), knowledge injection (Wang et al., 2020) or continual learning (Lin et al., 2020; Yu et al., 2024). A related PETL approach is LoRA (Hu et al., 2022) and its numerous extensions (Karimi Mahabadi et al., 2021; Dettmers et al., 2023; Kopiczko et al., 2023; Liu et al., 2024; Hao et al., 2024; Zhang et al., 2023), which model the update of the parameters using linear low-rank projections, merging the new weights with the pre-trained ones at inference. Other alternatives include Prompt Tuning (Lester et al., 2021; Shi & Lipani, 2023) and Prefix Tuning (Li & Liang, 2021; Jia et al., 2022) that append a trainable vector of tokens to the layer input. Solutions based on Side-Tuning (Zhang et al., 2020) allow to bypass the gradient propagation through the backbone, accelerating the training (Mercea et al., 2024; Gupta et al., 2024; Wang et al., 2023; Munkhdalai et al., 2024). These various PETL methods can be applied together using algorithms designed to select their optimal configurations (Mao et al., 2022; Zhang et al., 2022; Zhao et al., 2021). In this work, instead of introducing another PETL approach, we focus on the adapters, examining how their performance is impacted by their placement.

### 2.2 IMPORTANCE OF ADAPTER'S PLACEMENT

A key design choice in PETL is determining where to introduce additional computational capacity to the model. For adapters, common placement strategies include the sequential and parallel approaches. Sequential placement inserts the adapters between consecutive layers (Houlsby et al., 2019; Mahabadi et al., 2021; Lauscher et al., 2020), while the parallel placement adds them using a separate residual connection within a layer (He et al., 2022; Zhu et al., 2021; Chen et al., 2022; Jie & Deng, 2022). Although adapters are usually inserted in every module of a transformer, multiple works find it sufficient to apply them only after the Feed Forward blocks (Luo et al., 2023; Bapna et al., 2019; Pfeiffer et al., 2020a). Notably, Houlsby et al. (2019) demonstrated that adapters in lower network layers can be removed without a significant performance drop. Rücklé et al. (2020) confirmed this observation, proposing an iterative approach that removes the adapters during the fine-tuning from selected layers. In the context of Mixture of Experts, higher layers were observed to require more LoRA adapters (Gao et al., 2024). At the same time, a study by Chen et al. (2023a) on parameter allocation policies in PETL concluded that uniform parameter distribution generally performs best on the GLUE benchmark. These findings suggest a non-trivial relationship between adapter placement and performance, motivating our investigation. While previous studies focused on adapting the layers locally, we extend the adapter architecture to include long-range and recurrent low-rank projections. Our goal is to understand how the location, by itself, impacts the adapter performance, and whether this influence can be predicted *a priori* to the training.

### 2.3 Rank-Based Metrics in Assessing Learning Capabilities

Measures based on the rank, especially estimates of effective dimensionality of the activations, have been studied in previous research to assess the learning capacity of models. For instance, in Reinforcement Learning, a decline in the feature rank of a value network can be associated with a decreased performance (Kumar et al., 2020). Similarly, Lyle et al. (2022) suggest that a high feature rank is a necessary condition for learning progress. In transfer learning, a significant change in the rank can serve as an indicator for neuron growth or pruning within a layer during fine-tuning (Maile et al., 2023). Moreover, activation ranks tend to diminish with increasing network depth (Feng et al., 2022), which has been linked to a degraded performance of linear probe on out-of-distribution data (Masarczyk et al., 2024). Those observations underscore the value of exploring rank-based metrics that estimate the importance of the placement of new computational capacity in the network.

## 3 Background

### 3.1 Adapters

An adapter, $A_\phi$, is a small network with parameters $\phi$ that is injected into a pre-trained model (Houlsby et al., 2019; Rebuffi et al., 2017). Adapters are often used in transfer-learning, where the parameters of the pre-trained backbone remain frozen, and only the adapters weights are updated during fine-tuning.

We focus on the most common adapter architecture, which consists of a low-rank projection followed by a non-linear transformation in the bottleneck dimension (He et al., 2022; Chen et al., 2022):

$$A_\phi(x) = \alpha\big(\sigma(LN(x)\mathbf{W_{down}})\mathbf{W_{up}}\big), \tag{1}$$

where $\mathbf{W_{down}} \in \mathbb{R}^{d_{model} \times r}$, $\mathbf{W_{up}} \in \mathbb{R}^{r \times d_{model}}$, $d_{model}$ is the hidden dimension of the network, $\sigma$ represents the non-linear activation function, and the parameter $\alpha$ is a trainable scalar that adjusts the scale of the output of the adapter. The function $LN(\cdot)$ indicates an optional layer normalization performed on the inputs of the adapter. For all types of adapters used in this work, we include this Layer-Norm operation, as we observe that it increases the stability of the learning. The bottleneck dimension $r$ of the down projection is called the *rank* of an adapter, and usually satisfies $r \ll d_{model}$.

Consider a block of layers $F_\theta(x)$ of the pre-trained model (e.g. the Multi-Head Attention or the Feed Forward module of a transformer). An adapter can be placed in parallel to the computational flow of the network by applying:

$$x_{i+1} = x_i + F_\theta(x_i) + A_\phi(x_i), \tag{2}$$

where $x_i + F_\theta(x_i)$ represents the standard residual connection of the model. Such an approach is known as the *parallel adapter*, and was reported to perform better than the alternatives based on sequential addition (He et al., 2022; Zhu et al., 2021; Jia et al., 2022). Adapters are predominately used together with transformer-based architectures where they are injected at every block.

### 3.2 Datasets and Models

We consider transfer tasks for both image and text classification problems. We use the ViT-B/16 model (Dosovitskiy et al., 2021) pre-trained on Imagenet-21K (Deng et al., 2009) for the former, and the RoBERTa-Base model for the later (Liu et al., 2019). Following earlier research on the topic (He et al., 2022), we employ two text classification tasks from the GLUE benchmark (Wang et al., 2018): MNLI, and the considerably smaller SST2. When adapting vision transformers, we use the iNaturalist18 (Van Horn et al., 2018) and Places 365 (Zhou et al., 2017) datasets. Additionally, to assess few-shot transfer accuracy, we include three tasks from the VTAB-1K benchmark: Clevr-Count-1K, dSpr-Loc-1K and SVHN-1K (Zhai et al., 2019). The VTAB-1K datasets consist of 1,000 training examples, chosen to represent tasks that introduce a distribution shift compared to ImageNet. See Table 2 in the Appendix for the summary of the datasets.

### 3.3 Motivation: Adapter Placement Matters

We begin our investigation by evaluating the transfer performance of a *single* parallel adapter at different locations. To this end, we consider a transformer encoder architecture, where the adapter

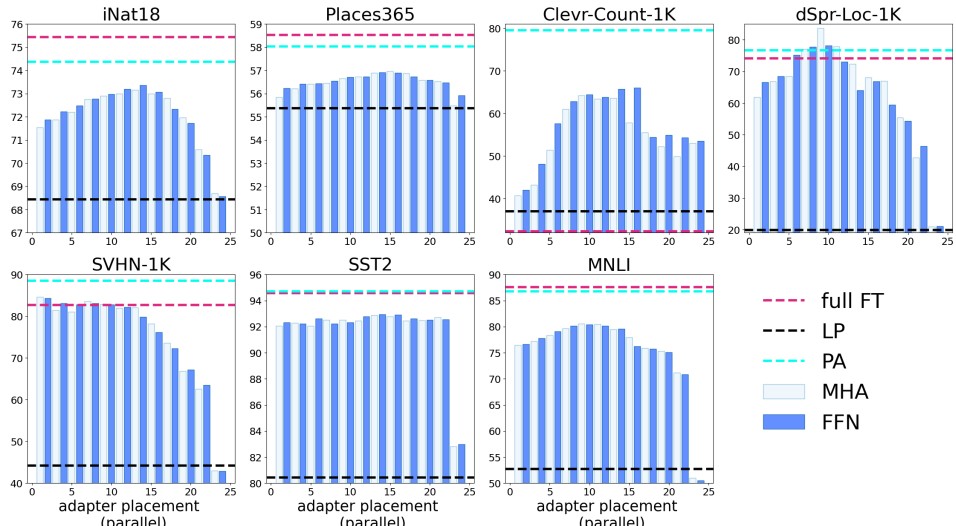

Figure 1: The test accuracy of a single parallel adapter for different placements. The dashed horizontal lines mark the performance of the full-fine-tuned model (pink), linear probe (black) and a setup with all 24 parallel adapters placed in every layer, both after the MHA and FFN module (cyan). The obtained results of the single adapters are affected both by the task and selected placement.

may be added in two places in each layer – between the Multi-Head Attention (MHA) module and the Feed Forward (FFN) module. Given a model with $L$ layers, this results in $n = 2L$ possible locations. Separately for each such location, we insert a single parallel adapter with layer norm (recall Equation 2), and fine-tune it on the transfer tasks from Section 3.2. As baselines, we incorporate the linear probe (LP), the Full-Fine Tuning (FT), and the standard adapter recipe with all 24 parallel adapters added simultaneously (PA).[1] We share the results in Figure 1.

We find that the performance of a single adapter, as measured by the test accuracy after fine-tuning, is greatly influenced by its placement. Furthermore, the best locations are task dependant, even when the same pre-trained model is used. For instance, for SVHN, placing adapters at earlier layers leads to best results, while for Clevr-Count the same strategy is the worst choice. This stark difference highlights the importance of tailoring the placements to each task. This motivates us to explore an extended search space for adapter placements, moving beyond common strategies to discover optimal configurations for each transfer task.

## 4 EXTENDED SEARCH SPACE FOR ADAPTERS

Connecting non-consecutive layers, especially given the importance of adapter placement, offers a promising opportunity to leverage non-local information for enhanced transfer performance. To explore this, we conceptualize the network as a graph where nodes correspond to distinct hidden representations, and edges define adapters. We describe this graphical representation using a transformer architecture in the following section.

### 4.1 ADAPTER GRAPH

Consider a transformer encoder with $L$ blocks. Each block computes two intermediate values that can be potentially altered by the adapter: the post-MHA hidden states, and the post-FFN hidden states. Including the input to the first block, we obtain a total of $n = 2L + 1$ hidden states.

The placement of an adapter can be represented as an edge $(i, j) \in E$ in the Graph $G(V, E)$, where the vertex set $|V| = n$ indexes the network hidden states. For block $l$, vertices $2l + 1$ and $2l + 2$ indicate the post-MHA and post-FNN hidden states, respectively.[2] The index zero corresponds to

---

[1]See Appendix A for further details on the training setup.

[2]Note that for any $l$, the point $2l + 2$ is also the input to layer $l + 1$.

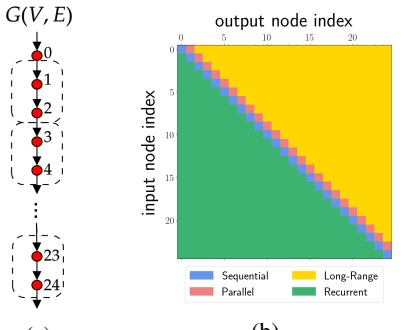 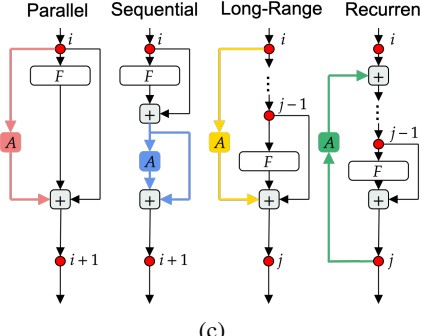

Figure 2: **(a)** The visualization of the connectivity graph $G(V, E)$ for a Transformer encoder network with $L = 12$ layers (resulting in $n = 25$ nodes). Each node corresponds to a hidden representation in an encoder block (denoted by the dashed lines), either after the MHA, or the FFN module. The node with the index zero represents the input to the encoder. **(b)** The adjacency matrix of graph $G(V, E)$ with marked search spaces for adapter placements. **(c)** The visualization of each studied adapter type. The block $F$ corresponds either to an MHA or FFN module.

the input to the first block. Below, we discuss how different graph edges define different adapter behaviours, and visualize them in Figure 2:

**Parallel Adapters** are represented by edges $(i, i + 1)$ and, following Equation 2, are defined using a separate residual connection: $x_{i+1} = x_i + F_{i+1}(x_i) + A(x_i)$.

**Sequential Adapters** correspond to edges $(i, i)$ and directly modify the hidden representations in place by applying $x_i = A(z_i) + z_i$ where $z_i = x_{i-1} + F_i(x_{i-1})$.

While various variants of parallel and sequential adapters have been extensively studied in the literature (Houlsby et al., 2019; Jia et al., 2022), they cover only a small fraction of all possible connections in the graph interpretation. For a graph with $n$ nodes, this fraction is equal to $(2n-1)/n^2$. For instance, in the ViT-B encoder with 12 layers more than $92\%$ of possible connectivity patterns remain unexplored. This poses a question whether other placements of the adapters, going beyond the diagonal and parallel edges, can further enhance the results. To investigate this, we define two new groups: the long-range adapters (being an extension of the parallel ones), and recurrent adapters:

**Long-Range Adapters** may skip multiple layers by extending the definition of parallel adapters to edges $(i, j)$ where $i < j$, resulting in hidden states given by $x_j = x_{j-1} + F_j(x_{j-1}) + A(x_i)$.

**Recurrent Adapters** are represented by edges $(i, j)$ where $i > j$. They connect later layers with lower layers in the model, resulting in a cycle in the computational flow. We consider a single recurrent step and define the recurrent adapters as following:

$$z_i = x_{i-1} + F_i(x_{i-1}) \tag{3}$$
$$z_j = x_j + A(z_i) \tag{4}$$
$$x_{j+1} = z_j + F_{j+1}(z_j) \tag{5}$$

In practice, we implement the above equation by doing two forward-passes though the network. In the first pass, we propagate the information along all the layers, as it would happen in a standard network without any adapters. Effectively, this allows us to simultaneously compute values $z_i$ from Equation 3 for all recurrent adapters. In the second pass, we feed the same input batch to the network and apply all non-recurrent adapters, as well as all the recurrent ones using Equations 4 and 5. Consequently, each block $F_{j+1}$ operates on the most recent hidden representation. When there are no recurrent edges, we only perform the second pass, recovering the behaviour of regular adapters.

Although recurrent adapters, independent of their numbers, require one extra forward pass during training, the gradients of the activations are calculated only for the second pass. Thus, assuming that

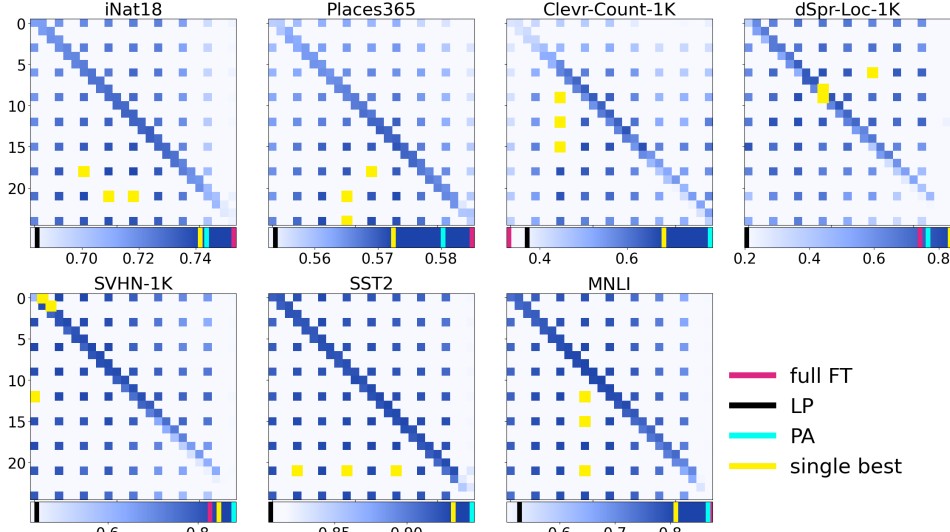

Figure 3: The test accuracy obtained for the single-adapter placement for various tasks. The y-axis (rows) represents the input node index $i$, while the x-axis (columns) corresponds to the output node index $j$. The vertical lines in the color bar indicate the performance of full fine-tuning (full FT), linear probe (LP), and parallel adapters (PA). With a bright yellow line we mark the performance of the best single adapter. The plots are normalized to the minimum and maximum performance of a single adapter for the given task. The three best performing adapters are also marked by yellow blocks in the plot (see Appendix C for top accuracy for each adapter type). Note that due to high computational cost, we subsample the adjacency matrix of all possible connections.

the cost of the forward and backward passes is equal[3], recurrent adapters increase the total training cost by at most 50%[4].

## 5 EXPERIMENTS

This section evaluates the benefits of the extended adapter search space. Mirroring the study in Section 3.3, we first examine the effects of adding a single adapter. We then explore the more practical scenario of incorporating multiple adapters simultaneously, seeking placement strategies that outperform the standard parallel configuration.

### 5.1 SINGLE ADAPTERS

We use the same experiment setup as the one from Section 3.3, but extend the search space of the adapter placement to the full connectivity matrix associated with the adapters graph from Section 4. Since evaluating the performance for all $n^2 = 625$ placements would be computationally expensive, we subsample the connectivity matrix with a stride of 3, i.e. we select all edges $(i, j)$ for which $i = 3l, j = 3k$ for $l, k \in \{0, \ldots, 8\}$. For each placement, we compute the mean test accuracy over 3 runs and present the results in Figure 3.

**Effectiveness of Recurrent Adapters**   Similar to our earlier experiment, we observe that final test performance varies greatly with the placement of the adapter. Although the sequential and parallel adapters (corresponding to the diagonal and upper diagonal entries in the matrices) usually perform reasonably well, they are often not among the top-3 placements, as indicated by the yellow squares in the plot. Top locations predominantly include recurrent adapters, as seen for the iNaturalist18, Places365, Clevr-Count, SST2 and MNLI datasets. Even long-range recurrent connections perform

---

[3]This is a reasonable assumption as the backward-pass cost is dominated by the activation gradients and adapter gradients are much cheaper in most settings.

[4]Note that the first pass would finish short if later activations are not used by any adapters.

Table 1: The maximum mean test accuracy over 100 sampled setups of placement of 24 adapters (Extended Max), compared with full fine-tuning, linear probe, parallel adapters and sequential adapters. We bold-out the best result in each column and underline the scores which fall within one-standard deviation range of the best result.

| | iNaturalist18 | Places 365 | Clevr-Count | SVHN | dSpr-Loc | MNLI | SST2 |
|---|---|---|---|---|---|---|---|
| Full FT | 75.43±0.29 | 58.53±0.05 | 32.32±0.18 | 82.65±0.45 | 74.10±0.71 | 87.59±0.09 | 94.56±0.35 |
| Linear Probe | 68.43±0.13 | 55.37±0.03 | 36.99±0.19 | 44.23±0.11 | 19.89±0.08 | 52.75±0.07 | 80.42±0.04 |
| Parallel Adapters | 74.38±0.05 | 58.03±0.06 | 79.57±1.92 | **88.53±0.63** | 76.63±0.71 | 86.77±0.21 | 94.69±0.23 |
| Sequential Adapters | 74.64±0.11 | 58.15±0.02 | **79.89±2.41** | 86.77±0.45 | 78.84±4.24 | 86.94±0.08 | 94.04±0.69 |
| Extended (MAX) | **75.62±0.20** | **58.26±0.08** | 77.87±2.30 | 88.06±0.70 | **95.63±1.12** | **86.96±0.24** | **95.00±0.25** |

very well (e.g., $(24, 6)$ in iNaturalist18 and Places365). This may be due to the benefit of using high-level information when adapting low-level modules. It's important to note that the success of recurrent adapters cannot be solely attributed to an increase in FLOPs; otherwise, the optimal placement would be at edge (24,0), which is clearly not the case.

**Power of a Single Adapter** In Figure 3, we observe that even a single, well-placed adapter can significantly improve performance compared to the linear probe. In some cases, such as iNaturalist18 or dSpr-Loc-1K, the obtained results are comparable to or better than the baseline with 24 parallel adapters. This demonstrates that a more strategic approach to adapter placement, rather than uniform application across all layers, can yield significant parameter efficiency gains.

**Correlation Between Transfer Tasks** Finally, we examine the relationship between the results obtained for different datasets. To this end, we calculate the correlations between the test accuracies of the sampled locations for each pair of datasets – see Figure 4. We observe low correlations, especially between the smaller VTAB-1K datasets. Consequently, apart from special cases (e.g. iNaturalist18 and Places365), the best placement for a given task is unlikely to transfer to another task. Therefore, any method of adapter placement selection needs to utilize the knowledge not only from the pre-trained model but also from the task itself.

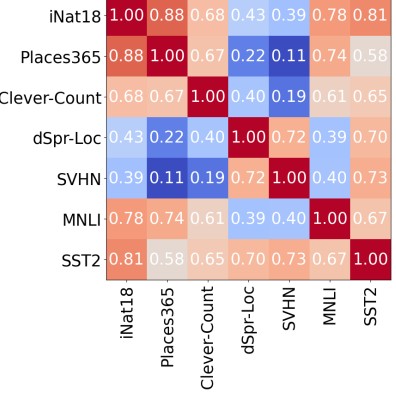

Figure 4: The spearman correlation between the test accuracies of adapters location for different datasets obtained using the data from Figure 3.

## 5.2 MULTIPLE ADAPTERS

In the previous section, we showed that the best performing single adapters often reside within the extended search space, which includes the recurrent and long-range adapters. In this section, we analyze the scenario of simultaneously adding multiple adapters to the network. Our aim is to demonstrate the *existence* of better placement assignments. We uniformly sample 100 different combinations of 24 adapters from the grid of $n^2 = 625$ placements. We compare the maximum test performance obtained through this random sampling (averaged over 3 runs) with parallel and sequential adapters baselines in Table 1.[5]

**Random Search Finds Better Adapter Placements** The best adapter combination found in the extended search space significantly improves over other adapter baselines for iNaturalist18, dSpr-Loc, and SST2 datasets, and surpasses the full fine-tuning accuracy. For the remaining datasets, the results are comparable to each other, falling within one-standard deviation range. Note that there are $C(625, 24) \sim 1.3e43$ possible combinations in the extended search space, out of which we only sample $1e2$. It is thus surprising that even with such a small sample size we are able to achieve substantial improvements.

---

[5]We use the maximum mean performance, since we are mainly interested in verifying the *existence* of a better set of locations for the given number of adapters. We study ways of identifying such sets efficiently in following sections.

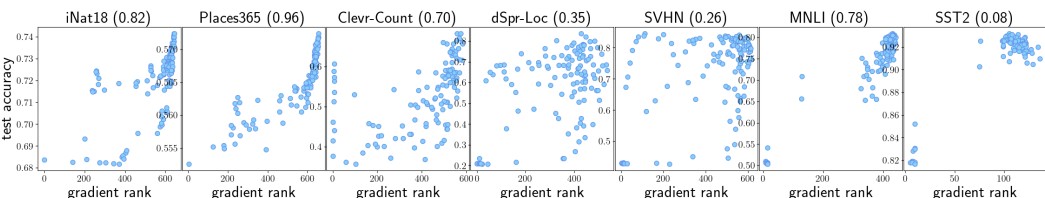

Figure 5: The test accuracy obtained for a given location versus the gradient rank of that location for the different datasets. We report the Spearman's correlation coefficient computed for each data (in brackets).

# 6 Towards identifying Best Adapter Placement

## 6.1 Gradient as a Predictor of Adapter's Performance

Our results so far reveal the existence of better adapter placements within the extended search space. We would like to efficiently identify such placements for different fine-tuning tasks *a priori* to training. To do so, we propose ranking individual adapter locations using a scoring function and selecting a group of them in a greedy manner. As shown in Figure 3, the usefulness of adapters varies within each row and column, indicating that the ranking should depend on both the input and output of an adapter.

Inspired by works on adaptive growth and pruning in neural networks, we examine the information potentially hidden within the gradients of the adapter's parameters, which naturally relies on the input and output nodes connected by the adapter. Intuitively, higher gradient magnitudes imply a larger change in the loss and, therefore, potentially faster learning. Consequently, gradient magnitude has often been used as an indicator of weight importance for pruning or growing parameters (Evci et al., 2022; 2020; Lee et al., 2018).

Consider a linear adapter at edge $(i, j)$ represented by projection $A_{i,j}(x_i) = x_i \mathbf{W_{i,j}}$. The adapter gradient is equal to the multiplication of the gradient of activations at node $j$ and the activations at input node $i$:

$$\frac{\partial L}{\partial \mathbf{W_{i,j}}} = x_i^T \left( \frac{\partial L}{\partial x_j} \right). \tag{6}$$

We can then define a scoring function $s : \{n\}^2 \to \mathbb{R}$

$$s(i, j) = f\left( \mathbb{E}_D \left[ \frac{\partial L(D)}{\partial \mathbf{W_{i,j}}} \right] \right), \tag{7}$$

where $s(i, j)$ is the score for an adapter connecting layers $i$ and $j$, calculated by applying a function $f$ (e.g., the norm) to the average gradient of the loss $L$ with respect to the adapter's weights $W_{i,j}$, computed over a batch of data $D$ from the transfer task.

The remaining component of the above definition is the choice of function $f(\cdot)$, which aggregates the gradient matrix into a single scalar. We investigate various options, including the common matrix norms and find that the rank of the gradient matrix shows the strongest correlation with single-adapter training loss and test accuracy. We discuss alternative functions considered and their correlations with the obtained accuracies in Appendix E. We hypothesize that the rank is a more robust indicator of transfer performance due to its scale-invariance. For example, a rank-one matrix can have a large Frobenius norm despite containing minimal information. However, its rank remains unchanged regardless of its scale.

Let us note that although the proposed rank-based metric is computed using linear adapters, we confirm that the rankings of the best nonlinear adapters strongly correlate with those of the linear adapters (see Appendix D). We use zero-initialized linear adapters for scoring, as they eliminate the need to compute intermediate activations in the bottleneck dimension and do not influence the model's forward pass.

We continue our study by calculating the numerical rank[6] of the adapter gradient using our transfer datasets and calculate the correlation between the computed score matrices and the test accuracies for each considered location. We present the results in Figure 5.

**The Rank of the Gradient is Indicative of Adapter Performance**   Both in the vision and text classification problems, we observe that the rank of the gradient correlates best for the more difficult, larger datasets like iNaturalist18, Places365 or MNLI. For the smaller tasks, the highest correlation is achieved on the Clevr-Count dataset, while the dSpr-Loc, SVHN, and SST2 obtain significantly lower correlations. Note that predicting the training dynamics and final transfer performance of a network at initialization is a challenging problem. It is therefore surprising that the rank of the gradient can serve as good predictor of the final test accuracy of an adapter, even if primarily for larger datasets. We now discuss how these findings can be incorporated into an algorithm for efficient identification of multiple adapter placements.

## 6.2   PLACEMENT SELECTION ALGORITHM

In the previous section we demonstrated that the rank score allows *a priori* identification of the optimal single adapter placement for selected datasets. However, in practice, we aim to choose an arbitrary number $N$ of locations. To this end, we propose a simple algorithm that uses a scoring function, such as that described in Equation 7, to rank locations and select the top performers.

Note that greedily picking the top-$N$ indices from the score matrix would predominantly return adapters that are placed within a localized neighborhood. Consequently, multiple adapters would modify the same input or contribute to the same output, introducing unnecessary redundancy and limiting exploration of the search space. To mitigate this, we propose the Gradient Gradient Adapters (GGA) algorithm, which discounts the scores for subsequent placements proportionally to the distance from the previously selected location.

**Gradient Guided Adapters (GGA)**   Given a desired number $N$ of adapters, the algorithm begins by calculating the scores for each possible location, creating the score matrix $s(i, j)$. After selecting the top position $(k, l)$, the score matrix is updated by discounting the scores of all entries proportionally to their distance from the chosen position, using $s(i, j) \odot d_{k,l}(i, j)$, with $d_{k,l}(i, j) = 1 - \gamma^{d_1((i,j),(l,k))}$, where $\gamma \in (0, 1)$, $d_1(\cdot, \cdot)$ is the $L_1$ distance, and $\odot$ denotes the element-wise multiplication. The role of the discounting is to discourage the algorithm from placing further capacity within the close proximity of previously selected adapters. See Appendix H for details on the Algorithm.

We compare the placements found by GGA with discounting ($\gamma = 0.6$) and without discounting ($\gamma = 0$, resulting in choosing the top-$k$ best scoring placements) to the random selection from Section 5.2 and two other baselines: the last-$k$ selection, and the first-$k$ selection. The last-$k$ baseline, introduced by Rücklé et al. (2020), places parallel adapters only in the final $k$ layers of the model. Conversely, the first-$k$ selection applies parallel adapters only to the $k$ layers closest to the input. We evaluate all methods on the iNaturalist18 and Places365 datasets across different adapter counts. Results are presented in Figure 6a and Figure 6b, respectively.

**GGA is Effective with Few Adapters**   We observe that GGA is especially effective when the number of added adapters is low. In particular, we can match the performance of 24 parallel adapters (PA) already with 3 adapters for iNaturalist18 and 12 for Places365. As the number of adapters increases, the effect of GGA diminishes, ultimately matching the performance of random selection for 24 placements. Moreover, we note that GGA with discounting surpasses the purely greedy algorithm (top-$k$) up to the point of approximately $N = 12$ adapters, after which both approaches perform similarly to random selection. In addition, we also find that the first-$k$ baseline is a better choice than last-$k$ when placing only a few adapters in the model, but this difference disappears when the number of adapters increases.

---

[6]We use the *srank* estimator defined as $r(A) = \arg\min_i \frac{\sum_{j=i}^{d} \lambda_j}{\sum_{j=1}^{d} \lambda_j} \leq \eta$, where $\lambda_j$ are the singular values of $A$ and $\eta$ is a thresholding parameter (Kumar et al., 2020).

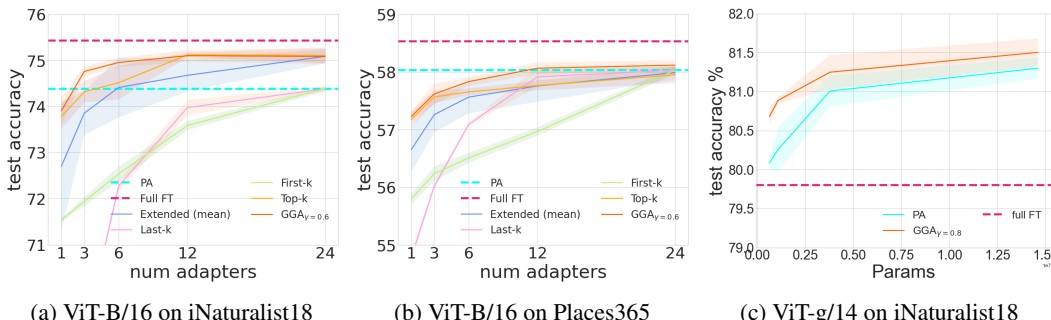

(a) ViT-B/16 on iNaturalist18     (b) ViT-B/16 on Places365     (c) ViT-g/14 on iNaturalist18

Figure 6: The validation accuracy on the **(a)** iNaturalist18 and **(b)** Places365 datasets obtained by finetuning ViT-B as a function of the number of used adapters for different placement selection algorithms. The dashed horizontal lines mark the performance of parallel adapters (PA) and full fine-tuning (full FT). For the GGA we present the algorithm both with discounting ($\gamma = 0.6$) and without discounting (Top-k) – see Appendix A for implementation details. **(c)** The test accuracy of fine-tuning adapters of varying rank (expressed as trainable parameter count on the x-axis) using the ViT-g/14 model and iNaturalist18 dataset.

**Varying Rank in Large-Scale Model** To demonstrate that GGA scales with the size of the model, we also consider the ViT-g/14 network (Zhai et al., 2022) fine-tuned on the iNaturalist18 dataset. The ViT-g/14 architecture has circa 1 billion parameters and $L = 40$ layers, leading to $n^2 = 6561$ possible adapter locations. From this vast search space we select $N = 40$ adapters using GGA and adjust their ranks to control the total number of added parameters. We compare GGA to the parallel adapters (PA) setting, where we inject the adapters only around the FFN blocks due to the large number of layers. Analyzing the results in Figure 6c, we observe that GGA consequently outperforms the baseline approach. The difference is the more visible, the less trainable parameters are used.

The results confirm the potential of the extended search space. Remarkably, even with a simple algorithm like GGA, the total number of adapters can be significantly reduced while still maintaining or exceeding the performance of parallel adapters.

## 7 CONCLUSIONS

This study examined the effect of adapter placement on transfer performance. We expanded the search space beyond the conventional parallel and sequential configurations by introducing long-range and recurrent adapters, allowing information to be updated along paths between any two blocks in a model. We tested this approach on tasks of varying modalities and difficulty, demonstrating that even random sampling from the extended search space yields adapter placements that surpass baseline methods. Furthermore, we found that the rank of a linear adapter's gradient is strongly correlated with its performance on most transfer tasks. This insight led to the development of a simple greedy algorithm for optimal adapter placement, which outperforms baselines in parameter-efficient fine-tuning. Our results underscore the importance of understanding how the placement of additional computational capacity influences fine-tuning performance and open new avenues for research into efficient identification of such placements.

**Limitations and Future Work** This study focused on adapter placement in transfer learning. Future research could investigate whether our findings extend to other PETL strategies, such as LoRA, Prefix-Tuning, and Side-Tuning. Recurrent adapters emerged as the most effective single-adapter type, suggesting further exploration of their potential and ways to adjust their computational cost. Additionally, we observed a strong correlation between adapter gradient rank and performance, though there were some exceptions. Studying the factors that influence this relationship could enhance our understanding of training dynamics in fine-tuned models and lead to more reliable performance metrics. Lastly, future work could aim to develop iterative placement algorithms, where gradients, ranks, and scores are recalculated after a set number of steps or each time a new adapter is added to the network.

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

## A   TRAINING REGIME

Throughout the paper we consider both image classification and text classification problems and focus on the transformer architecture. If not specified otherwise, for vision we always use the ViT-B/16 model (Dosovitskiy et al., 2021) which is pre-trained on the Image21K (Deng et al., 2009) dataset. For text, we use the RoBERTa-Base (Liu et al., 2019) model. Below we discuss the used datasets and the exact setups and training hyperparameters used for each conducted experiment.

### A.1   DATASETS

For vision problems we use two large datasets, iNaturalist18 (see "iNaturalist 2018 competition dataset") and Places365 (Zhou et al., 2017), as well as three small tasks selected from the VTAB-1K benchmark: the Clevr-Count, dSpr-Loc and SVHN, each with 1000 total training samples. In the case of iNaturalist18 and Places365 we adapt their "validation" splits as the test splits. We extract 1% of the training set as new validation set and perform any hyper-parameter selection based on the performance on that set. For the VTAB tasks, we use the standard test/train split. The hyper-parameter selection is performed by conducting experiment on a validation set formed from extracting 200 samples from the training set, following Zhai et al. (2019). For text, we use the MNLI and SST2 datasets from the GLUE benchmark (Wang et al., 2018). We select those datasets, as they have been used in other works investigating the adapters performance (He et al., 2022). We use the development splits as the test splits, and extract 10% of the training data as validation set. In all cases, after hyper-parameter selection, we optimize the model on the full training dataset (including the validation splits) and report the final test performance (see following sections).

### A.2   SINGLE ADAPTER PLACEMENT: FIGURE 1

The aim of the experiment is to capture what is the impact of the placement of single adapter on the performance of the network. We use three baseline methods: full fine-tuning (FT), linear probe (LP), and a setup including all 24 parallel adapters (PA). In the linear probe method, the backbone of the model is freezed, on top of which a trainable linear projection is added. In the all-adapters setup we use the standard parallel adapter as described in Section 3.1, which is placed in every layer, both after the multi-head attention (MHA) and feed-forward (FFN) blocks.

For the vision tasks we fine-tune the ViT model with the SGD optimizer with momentum 0.9 and weight decay 0.0 , using a batch size of 128 for the iNaturalist18 and Places365 datasets, and a batch size of 64 for the VTAB tasks. We use the cosine annealing scheduler for the learning rate. The base learning rate is selected by a hyperparameter search over the values of {1e-4, 1e-3, 1e-2, 1e-1}, averaging the validation performance over 3 seeds. In addition, for the VTAB task, we extend the hyperparameter search to a grid including different number of total training steps, investigating values from {500, 2000}. For the large datasets (iNaturalist18 and Places365) we train the models for 20000 steps. In both cases we use warm-ups with 500 steps, respectively. For all 24 parallel adapters (PA), we used a rank of 8 for the VTAB tasks, consistent with the configurations in Jie & Deng (2022) and Zhang et al. (2022). For iNaturalist18 we sweep the rank of the adapter over the values of {32, 128, 512}, and select rank 128, as it achieved the best performance-to-parameters trad-off. We adapt the same rank for Places365, due to the similarity of size and difficulty of those datasets.

For text tasks, we again sweep over the best learning rate from the set of {1e-5, 1e-4, 1e-3, 1e-2} for full-fine tuning and {1e-4, 1e-3, 1e-2, 1e-1} for linear probe and adapters. All other hyper-parameters follow the same setup as in He et al. (2022), where the training is performed using the Adam optimizer, with weight decay 0.1 and a polynomial learning rate scheduler, including a warm-up equal to 6% of the total training steps. The SST2 dataset is trained for 10 epochs. For the MNLI dataset, we find that fine-tuning over 3 epochs gives the same performance as after 10 epochs, and use the first number to reduce the training time.

In order to obtain the single-adapter performance for each of the 24 possible placements [7] we add one single adapter and then perform the training with the same hyper-parameters as the ones used for the all-adapter setup. For all methods, we report the average test accuracy over 3 runs.

---

[7] 12 layers, and within each one placement after MHA and one after FFN

Table 2: The summary of datasets used for transfer learning.

| dataset | train size | test size | validation split |
|---------|-----------|-----------|------------------|
| iNaturalist18 (Van Horn et al., 2018) | 437,513 | 24,426 | 1% of training |
| Places365 (Zhou et al., 2017) | 1,803,460 | 36,500 | 1% of training |
| MNLI (Williams et al., 2017) | 392,702 | 9796 | 10% of training set |
| SST2 (Socher et al., 2013) | 67,349 | 872 | 10% of training set |
| Clevr-Count-1K (Zhai et al., 2019) | 1,000 | 15000 | 20% of training set |
| dSpr-Loc-1K (Zhai et al., 2019) | 1,000 | 73728 | 20% of training set |
| SVHN-1K (Zhai et al., 2019) | 1,000 | 26032 | 20% of training set |

### A.3 EXTENDED SEARCH SPACE: FIGURE 3 AND TABLE 1

For the Extended Search Space experiments from Figure 3 we use the exact same configuration as for the single parallel adapter experiment, computing the adapter's output accordingly to equations introduced in Section4. The maximum performance over the random adapters selected from the extended search space is computed by first sampling 100 lists of 24 adapter locations. Next, for each of the seven datasets, we add the adapters represented by the list to the corresponding pre-trained model and fine-tune it using the same setup as described in Section A.2. For each list, we perform 3 runs and report the maximum average test accuracy.

### A.4 SELECTING THE AGGREGATION FUNCTION: TABLE 4

In order to compare different choices of the aggregated function $f(\cdot)$ we first pre-train the head of the model for 2.5% of all training steps, keeping all other adapter and backbone parameters fixed. Next, we calculate the gradient of the linear adapter and average its value over a batch of 512 examples. We then aggregate the result using common norms as described in Table 4. This results in a separate score matrix $s(i, j)$ for each of the norms, where a given entry contains the score computed for the corresponding placement. Those values are then used to compute the correlation with the test accuracies and training loss from the sub-sampled space from Figure 3. We select the rank norm for further consideration, as it resulted in the best training loss correlation (see Appendix E and Appendix G).

To compute the rank score matrices for the rest of the data we use the same procedure as for iNaturalist18, pre-training the head of the model for each of the datasets for 2.5% of all the training steps. We use batch size 512 to compute the scores for iNaturlaist18, Places365, MNLI and SST2. For the VTAB-tasks we use the entire training set from the first fold (i.e. 800 samples).

### A.5 ADAPTER SELECTION ALGORITHM: FIGURE 6A

In the experiment in Figure 6 we vary the number of total adapters over the set of {1, 3, 6, 12, 24} and compare GGA to baseline approaches of random selection, selecting first-k adapters, and selecting last-k adapters. We consider two variants of GGA: with and without discounting. When using the discounting, we perform a hyperparameter sweep over the discounting factor $\gamma \in \{0.5, 0.6, 0.7\}$, and select 0.6 as the best value. For all methods, we perform 3 runs and report the average test accuracy. All other parameters for training and metrics evaluation match the ones described in previous sections.

### A.6 LARGE SCALE MODEL: VIT-G

We use the ViT-g model to investigate how GGA performs on large scale architectures. The ViT-g transformer consist of $L = 40$ layers. Due to the size model, when analyzing the all-adapters setup, we decide to place the adapter module only to update the output of the FFN block, so that that the total number of placed adapters is equal to the total number of layers. We use GGA to select the same number of adapters from the extended search space. We compare the results of parallel adapters to that of GGA for increasing rank ($r \in \{4, 8, 32, 128\}$). We increase the discount $\gamma$ from 0.6 to 0.8, since in initial experiments with rank 32 the second value performed better on the validation set. Due to the significantly larger search space of 6561 positions stronger discounting is needed to enforce the algorithm to explore non-local connections. For each rank, we report the mean over 3 runs.

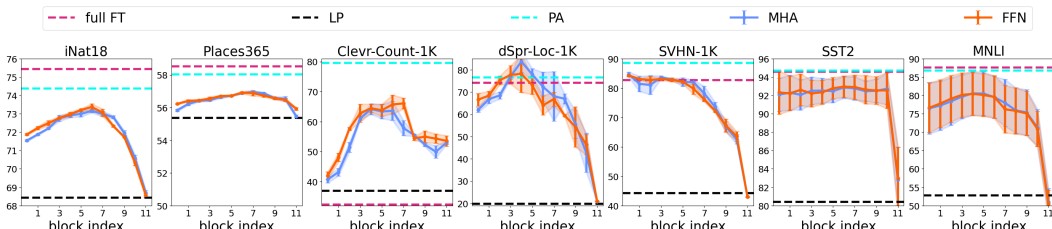

Figure 7: The comparison between the performance of a single adapter placed either after MHA or FFN within an encoder block. The index of the encoder block is presented on the x-axis. The dashed lines correspond to full fine-tuning (pink), linear probe (light orange) and the setup with 24 parallel adapters (black).

Table 3: The best test accuracy obtained for a single adapter for different adapter types. We bold-out the best results in each column and underline the scores which fall within one-standard deviation range from the best result.

| adapter type | iNat18 | places 365 | Clevr-Count | dSpr-Loc | SVHN | MNLI | SST2 |
|---|---|---|---|---|---|---|---|
| PA | 0.734±0.002 | 0.570±0.000 | 0.661±0.018 | **0.837±0.047** | **0.846±0.008** | 0.805±0.059 | 0.930±0.012 |
| SEQ | 0.734±0.001 | 0.571±0.001 | 0.594±0.011 | 0.819±0.032 | 0.829±0.008 | 0.803±0.058 | 0.927±0.015 |
| LR | 0.731±0.001 | 0.570±0.001 | 0.636±0.019 | 0.802±0.011 | 0.838±0.009 | 0.801±0.058 | 0.928±0.019 |
| REC | **0.742±0.002** | **0.572±0.001** | **0.683±0.015** | 0.799±0.040 | 0.841±0.015 | **0.810±0.061** | **0.932±0.015** |

## B  MHA OR FFN PARALLEL PLACEMENT

A common question addressed in adapter research is whether to insert the parallel adapter after the Multi-Head Attention or after the Feed-Forward layer in an encoder block (He et al., 2022). The latter approach is considered to be the more popular solution (Chen et al., 2022; Luo et al., 2023; Zhang et al., 2022; Pfeiffer et al., 2020a). In here, we revisit this question using the data gathered in the experiment from Section 3.3. We separately visualize the test accuracy obtained by the single FFN and MHA parallel-adapters for each block in the encoder architecture in Figure 7.

In most of the studied cases, the difference between the FFN and MHA placement is not significant. The exceptions are the Clevr-Count and dSpr-Loc datasets. For Clevr-Count using FFN adapters in the later stages of the model leads to better test accuracy. In contrast, the same blocks in the dSpr-Loc task per-from better with an MHA adapter. That being said, neither strategy emerges as universally predominately better over the other. In consequence, selecting the best *block* is more impactful than choosing between locations within that block.

## C  BEST ACCURACY OF A SINGLE ADAPTER FOR EACH ADAPTER TYPE

In addition to the Figure 3 from the main text, we also provide the single best performance obtained for different adapter types in Table 3. We observe that in all cases, apart from the dSpr-Loc dataset, the recurrent adapter is either significantly better, or comparable to the parallel adapters.

## D  LINEAR VS NON-LINEAR ADAPTERS

In the main paper we propose to look at the gradient of the linear adapter as a source of information indicating where to place the adapter module. Using the gradient of linear adapter simplifies the computations and does not require storing any additional data in the model, since it can be computed by a simple multiplication of gradients incoming to the output of the adapter and the intermediate activation. Such approach, however, assumes there is a monotonic relationship between the performance of linear and non-linear adapters.

To verify this claim, we compute the performance of single linear adapter for each of the placement studied in Figure 3 for the SST2 dataset. The linear adapter is implemented simply as a linear

projection. We use the same training regime as for the non-linear adapters, and average each result over 3 runs. We present the results in Figure 8.

Indeed, we observe that the performance of linear adapters correlates well with the nonlinear ones, both in terms of Pearson and Spearman's rank coefficients. In addition, it is evident that the linear adapters achieve very good results, outperforming their non-linear counterparts in most of the locations (note that the points in right plot of Figure 8 lie below the diagonal). We expect this to be the result of larger parameter count in the linear adapters - please note that they are implemented using a straightforward linear projection and do not contain a low-rank bottleneck.

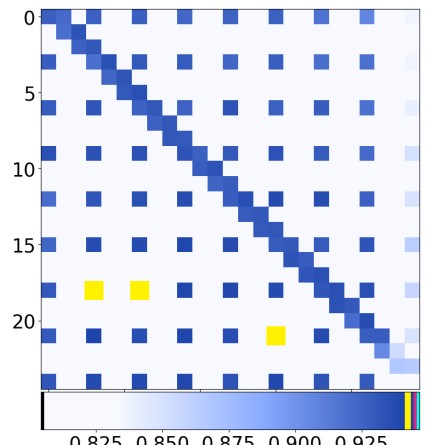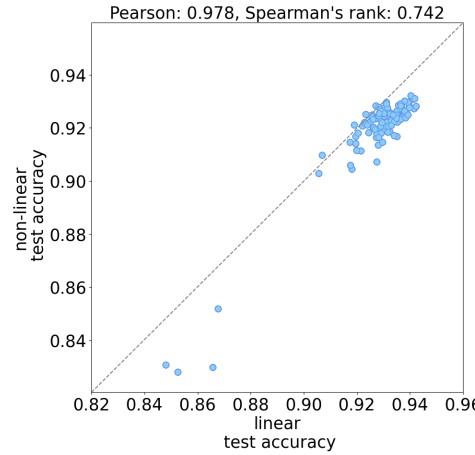

Figure 8: **Left:** The test accuracy for single *linear* adapters computed for the SST2 dataset. **Right:** The test accuracy of the non-linear adapters (y-axis) versus the test accuracy for the same placement obtained by a linear adapter (x-axis) on the SST2 dataset.

**Effect of activation for varying number of adapters**   In addition to the previous experiment, we also study how the change of the activation function in the bottleneck dimension of the adapter affects the performance. To this end we vary the number of adapters and place them only on the parallel positions (recall Figure 2). If the number of adapters is less then 24, we distribute the adapters locations uniformly, so that the distance between neighbouring adapters is the same. If only a single adapter is placed in the network, it is added on the first parallel position (edge 0-1). We compare the GeLU activation function with the ReLU activation and with using no activation (i.e. a linear adapter with low-rank projection). We plot the results in Figure 9. We observe that all variants of activation functions achieve similar performance. However, the non-linear activations seem to be more important when using a larger number of adapters.

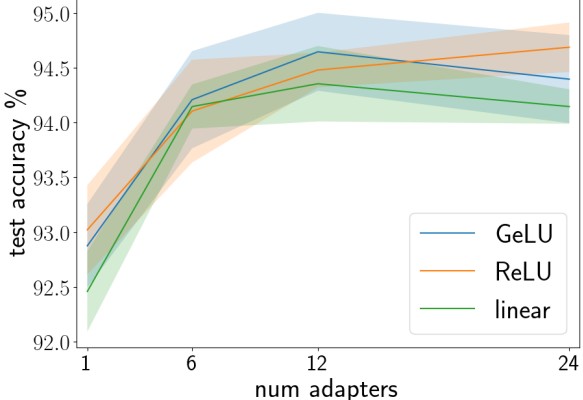

Figure 9: The test accuracy versus the number of adapters placed in the network for different activation functions of the adapter's module.

Table 4: Different forms of function $f(\cdot)$ with their definitions for a given matrix $A \in R^{d \times d}$. The symbol $\lambda_i$ refers to the $i$-th largest singular value from the SVD decomposition of matrix $A$, with $\lambda_{max} = \lambda_1$. The value $\eta$ is a thresholding hyperparameter (see Appendix G for tested values). We report the Spearman's correlation between the scores at each location with the training loss and test accuracy of the ViT-B/16 model fine-tuned on the iNaturalist18 dataset.

| name | notation | description | train loss correlation | test accuracy correlation |
|------|----------|-------------|------------------------|---------------------------|
| min. abs. column sum | $\|A\|_{-1}$ | $\min_j \sum_i \|A_{i,j}\|$ | 0.209687 | 0.562722 |
| max. abs. column sum | $\|A\|_1$ | $\max_j \sum_i \|A_{i,j}\|$ | 0.118575 | 0.002784 |
| frobenius norm | $\|A\|_{fro}$ | $\sqrt{\sum_{i,j} \|A_{i,j}\|^2}$ | 0.106483 | 0.421068 |
| spectral norm | $\|A\|_2$ | $\sqrt{\lambda_{max}}$ | 0.170864 | 0.392636 |
| nuclear norm | $\|A\|_{nuc}$ | $\sum_{i=0} \lambda_i$ | -0.158678 | 0.546193 |
| rank | $r(A)$ | $\arg\min_i \frac{\sum_{j=i}^{d} \lambda_j}{\sum_{j=1}^{d} \lambda_j} \leq \eta, \ \eta = 0.01$ | **-0.930429** | **0.823797** |

# E    DIFFERENT AGGREGATION FUNCTIONS

One key component of the score proposed in Section 6 is the choice of the function $f(\cdot)$. We investigate various transforms, including the minimum absolute column sum, the maximum absolute column sum norm, the Frobenius norm, the spectral norm, the nuclear norm, and the rank computed using the singular decomposition of the gradient – see Table 4 for summary. We compute the score matrices for each of those functions using the ViT-B/16 model fine-tuned on the iNaturalist18 dataset. Since the gradient at the beginning of the training may be heavily influenced by the initialization of the linear head of the model, we also find it beneficial to first pre-train the head for a number of $n_{steps}$, before computing the average gradients (see Appendix A and G for details). For each score matrix, we calculate the Spearman's rank correlation between the scores for each location and the train loss and test accuracy obtained for that location. We present the results in Table 4. In addition to the results from Table 4, we include the obtained score matrices and plot the relationship of the score matrix for each of the choice of function $f(\cdot)$ in Figure 10. It is visible that only the rank achieves promising correlation.

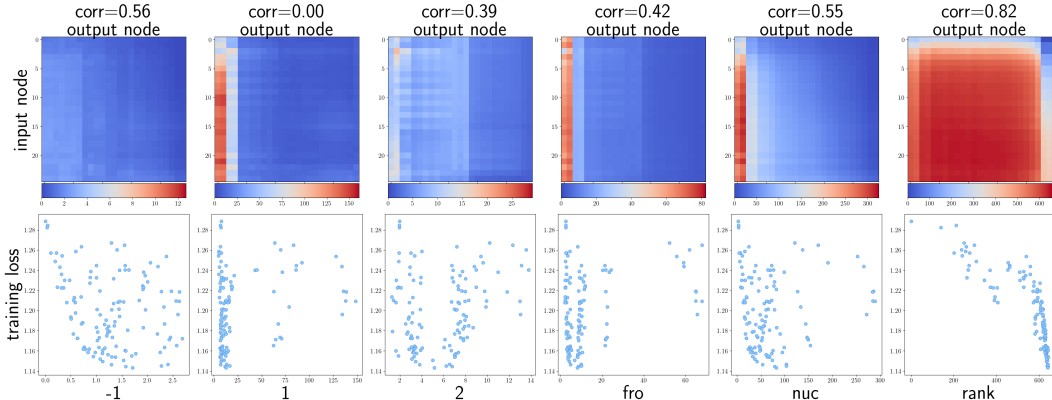

Figure 10: **Top:** The visualization of score matrices obtained for different choice of norm $f(\cdot)$. **Bottom:** The scatter plots depicting the relationship between the training loss for a given adapter location and the computed score. We report the Spearman's rank coefficient at the top of the plot.

## F CORRELATION WITH TRAINING LOSS

We study how the proposed metric correlates with the training loss for the various datasets in Figure 11. As expected, similarly to the correlation with the test accuracy, the placements with the largest gradient rank have also the smallest training loss. In addition, for some datasets, like SVHN and Clevr-Count, we observe that the few locations with lowest ranks and relatively high test accuracy from Figure 5 (main text) have worse training loss than the locations with higher gradients, indicating that some of the discrepancy in the computed metric may be due to generalization problems. To further study this claim, we compute the correlation between the final test accuracy and training loss for different datasets, and visualize it in Figure 12. We observe that for some tasks, the low training loss does not necessarily guarantee improved test accuracy (consider SST2 with correlation of only 0.56 or the Clever-Count with correlation 0.78), which may potentially affect the workings the effectiveness of the rank metric.

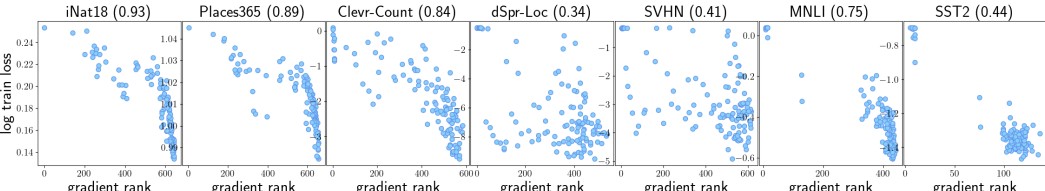

Figure 11: The logarithm of the training loss versus the computed gradient for different datasets.

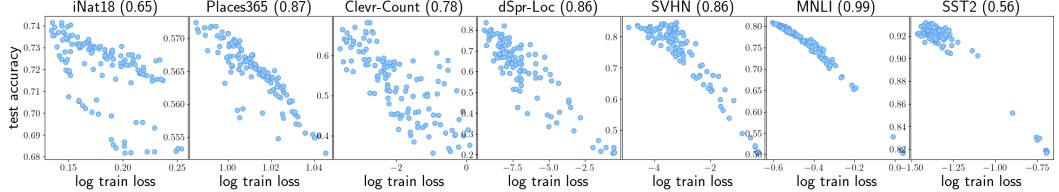

Figure 12: The final test accuracy obtained for a given location versus the final logarithm of the training loss for the same location computed for various datasets.

## G EFFECT OF RANK THRESHOLD AND NUMBER OF PRE-TRAINING STEPS ON THE METRIC SCORE.

In this section we study how the threshold hyperparameter $\eta$ and the number of pretraining steps of the head $n_{steps}$ affect the correlation computed between the rank of the gradient and the performance of adapters. We present the results in Figure 13. In general, it is clear that the longer the head of the model is pretrained in isolation, the better the correlation.

## H THE GGA ALGORITHM

The GGA is a simple score-based approach, that given a score-matrix and required number of adapters $N$, returns the top-$N$ placements with highest scores. Additionally, as mentioned in the main text, we decide to include a heuristic that discourages the algorithm for repeatedly selecting adapters from the same neighbourhood. The scores of placements in the vicinity of the last picked connection are reduced by multiplying them by the factor:

$$d_{k,l}(i,j) = 1 - \gamma^{d_1((i,j),(l,k))},$$

where $d_1$ is the L1 distance, and $(l,k)$ is the last picked edge. We refer to $d_{k,l}(i,j)$ as the *discount matrix* for edge $(i,j)$. Note that we prefer the choice of the $L_1$ distance in the computation of $d_{k,l}(i,j)$ over the $L_2$ distance, since we we would like to penalize placements in the same rows or columns more than those on the diagonal directions. The strength of the discounting is determined

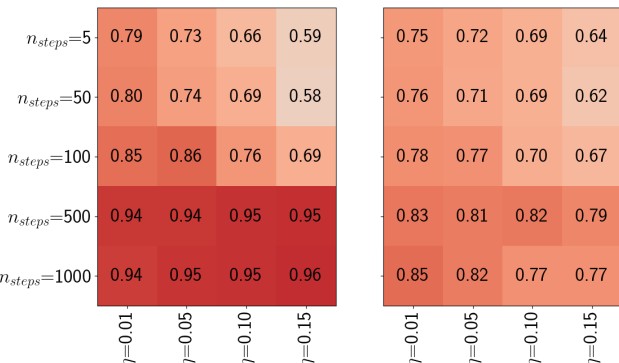

| | $\eta=0.01$ | $\eta=0.05$ | $\eta=0.10$ | $\eta=0.15$ | | $\eta=0.01$ | $\eta=0.05$ | $\eta=0.10$ | $\eta=0.15$ |
|---|---|---|---|---|---|---|---|---|---|
| $n_{steps}=5$ | 0.79 | 0.73 | 0.66 | 0.59 | | 0.75 | 0.72 | 0.69 | 0.64 |
| $n_{steps}=50$ | 0.80 | 0.74 | 0.69 | 0.58 | | 0.76 | 0.71 | 0.69 | 0.62 |
| $n_{steps}=100$ | 0.85 | 0.86 | 0.76 | 0.69 | | 0.78 | 0.77 | 0.70 | 0.67 |
| $n_{steps}=500$ | 0.94 | 0.94 | 0.95 | 0.95 | | 0.83 | 0.81 | 0.82 | 0.79 |
| $n_{steps}=1000$ | 0.94 | 0.95 | 0.95 | 0.96 | | 0.85 | 0.82 | 0.77 | 0.77 |

Figure 13: **Left:** The correlation between the training loss and the computed rank of the gradient for various numbers of pretraining steps (rows) and thresholds $\eta$ (columns) obtained for the iNaturalist18 dataset. **Right:** The same as left, but the correlation is computed between the test accuracy and the rank of the gradient.

---

**Algorithm 1** Gradient Guided Adapters (GGA)

---

**Require:** Model $M$, Number of adapters $N$, data $D_K$, batch size $b$ discount factor $\gamma \in (0,1)$, number of head pretraining steps $n_{steps}$. The notation $d_1$ indicates the L1 distance.
1: Pretrain the classifier head of $M$ for $n_{steps}$
2: $s(i,j) \leftarrow r(\frac{1}{b}\sum_i^b \frac{\partial L(D_K^{(i)})}{\partial \mathbf{W_{i,j}}})$
3: edges = []
4: **for** $i = 1$ **to** $N$ **do**
5: $\quad (k,l) \leftarrow argmax_{(i,j)}\, s(i,j)$
6: $\quad$ edges.add($(k,l)$)
7: $\quad d_{k,l}(i,j) = 1 - \gamma^{d_1((i,j),(l,k))}$
8: $\quad s(i,j) \leftarrow s(i,j) \odot d_{k,l}(i,j)$ #element-wise multiplication
9: **end for**
10: **return** edges

---

by the choice of the $\gamma$ hyperparameter. After the update, the new matrix $\hat{s}(i,j)$ is used for the next selection and the procedure is repeated until we choose all $N$ placements. We summarize this method in Algorithm 1. In Figure 14 we also depict an example of the discounting matrix and visualize the end state of a score matrix used when selecting $N = 24$ adapters.

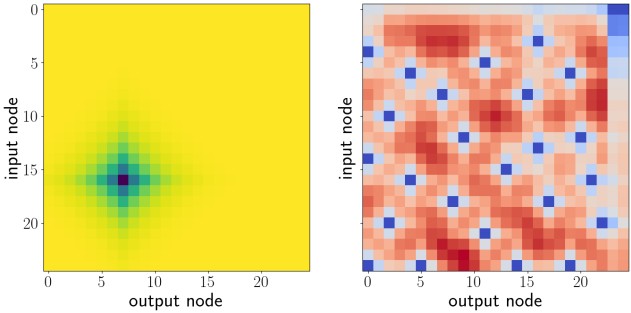

Figure 14: **Left**: An example of a discount matrix $d_{k,l}(i,j)$ computed for indices $(i,j) = (16,7)$ using discount factor $\gamma = 0.6$. **Right:** The visualization of the score matrix after placing 24 adapters using GGA.

Note that although in line 2 of the algorithm we use the rank of the gradient as the score function, the algorithm itself can work with arbitrary score matrices. We focus on the rank of the gradient of the linear adapters due to its promising correlation with the training loss.

## I  SINGLE ADAPTER VERSUS MULTIPLE PARALLEL ADAPTERS

In this section we analyze how a *single* adapter compares with the standard setup of placing parallel adapters in every block of the encoder. For the latter, we use the method of placing adapters only between the Feed-Forward module of the transformer, as suggested, e.g. in He et al. (2022). We manipulate the rank of the single adapter so that the number of parameters is approximately the same in both scenarios. This gives a total of 12 adapters with rank $r$ for the *all parallel* setup versus one adapter with rank $r * 12$ in the *single adapter* setup. We fine-tune the new pa-

Table 5: The test accuracy of fine-tuning a single adapter, versus fine-tuning 12 adapters placed in the FFN modules of a transformer, obtained for different ranks. The data was obtained by training on the iNaturalist18 dataset using the ViT-B/16 model.

| rank | all parallel (FFN) | single (best) |
|------|--------------------|--------------|
| 16   | 74.1               | 74.1         |
| 8    | 74.0               | 74.2         |
| 4    | 73.8               | 73.9         |
| 2    | 73.5               | 73.8         |

rameters on the iNaturalist18 dataset, varying the values of the rank. For the single adapter, we test all the sub-sampled placements (recall the experiment from Figure 3), and report the best test accuracy among all the locations. We present the results in Table 5. We observe that the single adapter from the extended search space, if well placed, is already able to match or marginally outperform the standard setup with multiple FFN adapters.

## J  FULL MATRIX

For the SST2 dataset, we also explored the scenario of adding all $N^2 = 625$ adapters simultaneously and compared the resulting test accuracy to that achieved by the best random selection of adapters identified in the experiment from Table 1. The results, shown in Figure 15, clearly demonstrate that simply increasing the number of adapters does not enhance performance. In the next section, we examine a related approach where a sparsity constraint is applied to the full adapter connectivity, followed by pruning adapters deemed unnecessary.

## K  COMPUTATIONAL COST OF GGA AND THE ADAPTERS FROM THE EXTENDED SEARCH SPACE

**GGA score computations**  Calculating the GGA scores for a network with $N^2$ possible placements involves computing the gradients from Equation 6 and evaluating Equation 7. For a transformer architecture, the FLOP cost of this is given by:

$$F_{score} = D_b * N^2(2n_{seq} - 1)d_{model}^2$$
$$+ (D_b N^2 d_{model}^2 + N^2 d_{model}^2),$$

where $D_b$ is the batch size used for the estimation of Equation 7, $n_{seq}$ is the sequence length and $d_{model}$ is the hidden size of the transformer. For example, for the ViT-16/B model from Figure 6 ($n_{s}eq = 197$, $d_{model} = 768$, $N^2 = 625$, $D_b = 512$) we have $F_{score} \approx 74.3$ TFLOPS. To compute the gradients in Equation 6, one forward and backward pass for a batch of size $D_b = 512$ is required. Assuming the forward

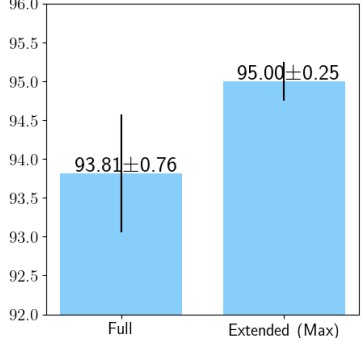

Figure 15: The performance of a model with all $N^2 = 625$ adapters in comparison to the best results obtained by the random search for $N = 24$ adapters on the SST2 dataset.

Table 6: The train loss for the experiment from Table 1.

| | iNaturalist18 | Places 365 | Clevr-Count | SVHN | dSpr-Loc | MNLI | SST2 |
|---|---|---|---|---|---|---|---|
| Full FT | 0.8602±0.0109 | 2.5379±0.0032 | 0.0000±0.0000 | 0.0127±0.0126 | 0.0001±0.0001 | 0.3658±0.0011 | 0.1136±0.0020 |
| Linear Probe | 1.3619±0.0111 | 2.8377±0.0025 | 0.6231±0.0030 | 0.7552±0.0030 | 1.5156±0.0036 | 1.0635±0.0001 | 0.5451±0.0002 |
| Parallel Adapters | 1.0033±0.0069 | 2.5876±0.0007 | 0.0003±0.0001 | 0.0036±0.0041 | 0.0001±0.0000 | 0.4448±0.0007 | 0.1505±0.0004 |
| Sequential Adapters | 1.0098±0.0052 | 2.5737±0.0037 | 0.0000±0.0000 | 0.0001±0.0000 | 0.0002±0.0001 | 0.4403±0.0009 | 0.1505±0.0003 |
| Extended (MAX) | 1.0189±0.0048 | 2.5725±0.0029 | 0.0004±0.0001 | 0.0031±0.0041 | 0.0000±0.0000 | 0.4363±0.0011 | 0.1570±0.0006 |

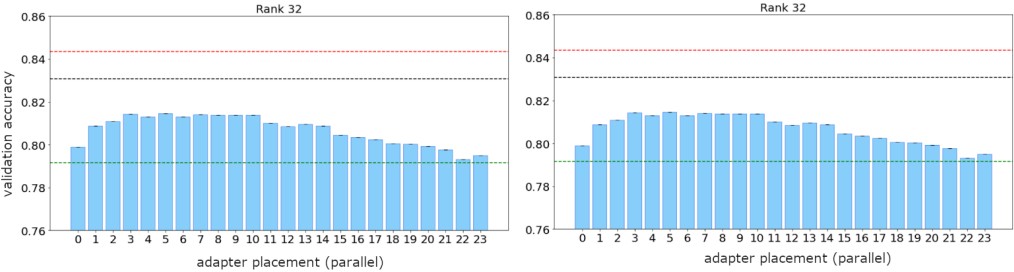

Figure 16: The validation accuracy on the ImageNet dataset. The red, black and green horizontal lines correspond to the performance of full fine-tuning (FT), 24 parallel adapters (PA) and linear probe, respectively.

pass for a single sample costs 34.9 GFLOPS[8] and that the backward pass has a similar cost (since we do not need to compute the gradients for the parameters of the vanilla ViT), the total FLOP cost is:

$$F_{GGA} = 512 * 2 * 34.9 + 74300 \approx 110.0 \text{ TFLOPS}$$

For comparison, a forward pass of ViT-16/B model with batch size 128 is approximately $128 * 34.9 \approx 4.5$ TFLOPS. Since we perform 20000 iterations during the training (amounting to a total 90 PFLOPS, and this is considering only the forward passes, without the cost of backpropagation) $F_{GGA}$ introduces only a negligible (0.12%) overhead.

**Parallel Adapters Complexity** The computational cost of a forward pass for Parallel Adapters (implemented by Equation 1) during inference in a transformer architecture can be expressed as:

$$F_A = n_{adapters} F_{adapter} \tag{8}$$
$$= n_{adapters}(F_{LN} + F_{down} + F_{\sigma} + F_{up} + F_{alpha} + F_{add}) \tag{9}$$
$$= 9d_{model} * n_{seq} + 4d_{model}n_{seq}r + 4n_{seq}r + n_{seq}d_{model} \tag{10}$$

where, $r$ is the rank of the adapter and $\sigma$ is a GeLU activation (5 FLOPS). For a setup with $r = 8$, $n_{adapters} = 24$ and a batch size of 128, this amounts to 15.4 GFLOPS, which is negligible compared to the 4.5 TFLOPS required for a single forward pass of the ViT-16/B model.

**Long-range and Recurrent Adapters** The Long-Range adapters are a natural extension of the parallel construction and hence they do not introduce any FLOPS overhead in comparison to the baseline PA approach. As long as there are no recurrent adapters in the network, the parameter count and FLOPs used by the adapters from the extended search space will be the same as for the baseline Parallel Adapters setup.

The introduction of the recurrent adapters introduces an additional cost caused by our implementation of two passes through the network. The first pass is need to propagate the $z_i$ values (Equation 3)

---

[8] $F_{ViT-B/16} \approx 12 * F_{layer}$, where $F_{layer} = F_{mha} + F_{ffn}$. The FLOPS of the FFN module are dominated by the cost of matrix projections, resulting in $F_{ffn} \approx n_{seq}(2d_{model} - 1)d_{ffn} + n_{seq}(2d_{ffn} - 1)d_{model}$, where $d_{ffn}$ is the size of the hidden dimension of the FFN block. For the attention, we will have $F_{mha} \approx (F_{proj} + F_{QK^T} + F_{values})h + F_{up}$, where $h$ is the number of heads, leading to $F_{mha} \approx 3n_{seq}(2d_{model} - 1)d_k h + n_{seq}^2 2d_k h + n_{seq}(2n_{seq} - 1)d_k h + n_{seq}(2d_k h - 1)d_{model}$. Substituting $n_{seq} = 197$, $d_{model} = 768$, $h = 12$, $d_k = 64$, $d_{ffn} = 4d_{model}$ we arrive at $F_{ViT-B/16} \approx 34.9$ GFLOPS.

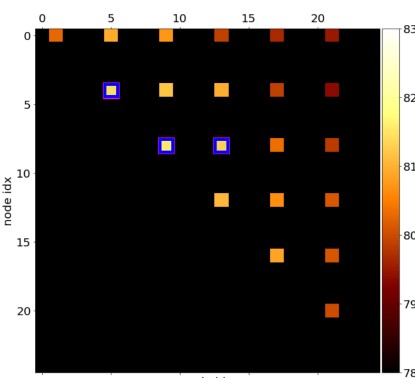

Figure 17: The validation accuracy obtained for the single-adapter placement for the ImageNet dataset. The y-axis (rows) represent the input node index $i$, while the x-axis (columns) correspond to the output node index $j$.

Table 7: The test accuracy on the SST2 datasets obtained by sampling 12 non-recurrent adapters from the Extended Search Space (reported results is the the best of 20 samples), in comparison to the *Last-k* and *First-k* strategies, as well as the performance of Parallel Adapters and best random sampling with recurrent adapters form Table 1. All results were averaged over 3 runs.

| Method | Accuracy | Number of Adapters |
|---|---|---|
| Extended - w/o recurrent (MAX) | **94.71±0.37** | 12 |
| Last-k | 93.50±0.44 | 12 |
| First-k | 93.92±0.50 | 12 |
| Extended - w/ recurrent (MAX) | **95.00±0.25** | 24 |
| Parallel Adapters | 94.69±0.23 | 24 |

simultaneously for all recurrent adapters. This pass may end earlier, since we only need to compute $z_i$ up to the index $i$ which is the maximal value such that $(i, j)$ is in the adapter set and $i > j$. If $i = 24$, then the resulting cost is equal to the forward pass of a network with no adapters. Note, however, that the gradients computed for the optimization of the adapter's parameters are only calculated in the second pass. Assuming that the backward pass has roughly the same cost in FLOPS as the forward pass, this results in $F_{total} = 2F_{forward} + F_{backward} = 3F_{forward}$. This represents at most a 50% increase compared to setups without recurrent adapters (for which $F_{total} = 2F_{forward}$).

## L    IMAGENET EXPERIMENTS

In the scope of our research we have also conducted preliminary experiments on the ImageNet dataset using the ViT-B/16 model. In particular, we considered the investigation from Figure 1, where we compared the validation accuracy of a single parallel adapter for different placements. The results are reported in Figure 16. We can observe that the performance initially increases together with the index of the layer, but then deteriorates the closer the adapter is to the output of the network. This behaviour is also similar to the one observed on iNaturalist18 and Places365 in Figure 1. In addition, we have also investigated a sub-sampled selection of long-range connections form the extended search space. The results are presented in Figure 17. We find that the best placement are most distributed close to the diagonal and are slightly shifted towards the beginning of the network.

## M    TRANSFERING THE BEST PLACEMENTS

We conduct an experiment in which we apply the best random placements (Extended Max from Table 1) identified for the SST2 to the MNLI task, and vice-versa. We compare the obtained results (averaged over 3 runs), with the performance reported in Table 1), where the placements are tailored

Table 8: The test accuracy on SST2 and MNLI dataset for the best result found by the random sampling from Table 1 (column "Tailored"), and the accuracy obtained by using the best placement of SST2 on MNLI, and vice-versa (column "Transfer"). The difference between the mean test accuracy for those approaches is reported in the last column of the table ("Delta").

| Dataset | Tailored (Extended Max) | Transfer | Delta: Tailored-Transfer |
|---------|-------------------------|----------|--------------------------|
| SST2 | $95.00\pm0.25$ | $94.48\pm0.09$ | 0.52 |
| MNLI | $86.96\pm0.24$ | $86.72\pm0.03$ | 0.24 |

to each task. We observe that such approach gives reasonably good performance, roughly matching the results of the Parallel Adapters method, but falling short of the performance achieved by using the best placements tailored to each task.

