# OpenReview forum: "Towards Optimal Adapter Placement for Efficient Transfer Learning"
_ICLR.cc/2025/Conference — Submitted to ICLR 2025_

### Official Review · Reviewer_Dkam · 2024-10-22

**Soundness:** 4
**Presentation:** 4
**Contribution:** 3
**Rating:** 6
**Confidence:** 4

**Summary:**

This paper investigates optimal adapter placement strategies in Parameter Efficient Transfer Learning (PETL) for deep neural networks. The authors introduce an extended search space for adapter connections, including long-range and recurrent adapters, which allows the adapters to exploit non-local information. They demonstrate that the placement of adapters is critical for the overall model performance and that strategically placed adapters can outperform standard uniform adapter placements. Additionally, they propose a gradient rank-based metric to identify optimal adapter locations efficiently.

**Strengths:**

1. The authors introduced a novel extension of the adapter placement search space, including long-range and recurrent adapters, which go beyond traditional uniform placement. This provides more flexibility and improves fine-tuning results.

2. The proposed gradient rank metric offers a practical and computationally efficient method for predicting the best adapter placements a priori, significantly reducing the computational cost of fine-tuning large models.

3.  The method shows that even with a small number of well-placed adapters, performance can match or exceed the standard uniform approach, highlighting the method's efficiency.

**Weaknesses:**

1. **Dataset Limitation**: While the paper demonstrates the method's effectiveness on several datasets, the reviewer is curious about its performance on larger and more diverse datasets, such as ImageNet. Testing on such datasets would better illustrate the method's scalability and generalization capabilities.

2. **Adapter Complexity**: Introducing long-range and recurrent adapters increases overall model complexity. However, the paper does not fully explore the trade-offs between this complexity and the performance gains. The reviewer is curious if the added complexity of long-range and recurrent adapters could lead to instability during training or inference, particularly in large-scale applications.

3. **Task-Specific Adapter Placement**: The method's reliance on task-specific adapter placement strategies may limit its practicality for general-purpose applications. The reviewer wonders if there is potential to reduce the task-specific nature of adapter placement, making the method more widely applicable without requiring custom placement strategies for each task.

**Questions:**

see weakness above

---

> ### Author Response · Authors · 2024-11-20
> **Response to Reviewer Dkam**
>
> We thank the Reviewer for the evaluation of our work and are happy to see that the Reviewer recognized the extended search space as a novel contribution that enhances flexibility and improves fine-tuning results. We are also glad to hear that the construction of the gradient rank metric was appreciated. Overall, we are grateful for the positive feedback. Below, we address the concerns raised by the Reviewer:
>
> > the reviewer is curious about its performance on larger and more diverse datasets, such as ImageNet.
>
> The datasets that we have selected for the empirical evaluation were based on those used in related studies on transfer learning and parameter-efficient fine-tuning (e.g., iNaturalist18 and Places were used in [1], VTAB in [2,3], SST2/MNLI in [4]). Following reviewers suggestion, we conducted some experiments on transferring models pretrained on Imagenet-21K to the ImageNet-1K, exploring the placement of a single parallel adapter and the influence of long-range connections (see the new Appendix L). Our results match the results presented for iNaturalist18 and Places365 in Figure 1, where we observe adapters added around middle blocks to achieve higher transfer accuracy.
>
> > The reviewer is curious if the added complexity of long-range and recurrent adapters could lead to instability during training or inference, particularly in large-scale applications
>
> Each adapter includes a Layer-Normalization module. Additionally, standard transformer architectures also incorporate such normalization layers. As a result, the inputs are always normalized, regardless of the structure of the adapter connectivity graph. We also employ learnable scale parameters (as described in Equation 1), which are initialized to zero to ensure stability. During experiments conducted in the Extended Search Space, we found that training hyperparameters, such as the learning rate, could be reused from the optimal configuration for Parallel Adapters. Even when running experiments with 40 adapters in the large-scale ViT-g/14 model, no stability issues were observed, as evidenced by the standard deviations shown in Figure 6c.
>
> > The reviewer wonders if there is potential to reduce the task-specific nature of adapter placement, making the method more widely applicable without requiring custom placement strategies for each task.
>
> Our experiments show that the best adapter placements are clearly task-dependent (consider the difference between e.g. Clevr-Count and SVHN in Figures 1 and 3). Therefore, the same set of adapters for every dataset and modality won’t in general be optimal. Nevertheless, we observe that the best adapter placements can transfer to some extent (recall Figure 4 where, for instance, correlation between best placements for iNaturalist18 and Places365 reaches 0.88). To further explore the idea of transition, we have conducted an experiment where the best random placement (Extended Max from Table 1) identified for SST2 was applied to the MNLI task, and vice versa. The results are promising, yielding test accuracies of $86.72 (-0.24)% $ for MNLI and $94.48 (-0.52)$ for SST2, where the performance reduction in comparison to the results for Extended Max (Table 1) is given in parenthesis. While these results roughly match the performance of PA, they fall short of the performance achieved by using the best placements tailored to each task. We added this experiment to the Appendix (Table 8).
>
> We once again thank the Reviewer for taking the time to evaluate our work and for the positive feedback. We believe we have addressed the raised concerns and remain available for any further clarifications if needed. Meanwhile, we kindly request the Reviewer to reassess our work in light of our responses.
>
> *References*
>
> [1] Mercea, Otniel-Bogdan, et al. "Time-Memory-and Parameter-Efficient Visual Adaptation." Proceedings of the IEEE/CVF Conference on Computer Vision and Pattern Recognition. 2024.
> [2] Jie, Shibo, and Zhi-Hong Deng. "Convolutional bypasses are better vision transformer adapters." arXiv preprint arXiv:2207.07039 (2022).
> [3] Zhang, Yuanhan, Kaiyang Zhou, and Ziwei Liu. "Neural prompt search." arXiv preprint arXiv:2206.04673 (2022).
> [4] He, Junxian, et al. "Towards a unified view of parameter-efficient transfer learning." arXiv preprint arXiv:2110.04366 (2021).

---

### Official Review · Reviewer_BPzc · 2024-11-03

**Soundness:** 2
**Presentation:** 2
**Contribution:** 2
**Rating:** 3
**Confidence:** 4

**Summary:**

This paper investigates the impact of adapter placement within pre-trained neural networks on their transfer learning performance. Besides, the paper introduces an extended search space for adapter placement, including long-range and recurrent adapters, and proposes a method to identify optimal placements using gradient rank as a predictor of adapter performance. The experiments across datasets and model scales demonstrate that strategically placed adapters can match or exceed the performance of uniformly distributed adapters.

**Strengths:**

* The motivation is clear and the adapter placement is worth researching.
* The proposed Gradient Guided Adapters (GGA) algorithm is interesting.

**Weaknesses:**

* The recurrent adapters require two forward processes, further deteriorating the efficiency of pre-trained models.
* Results in table 1 lack the comparison with related works.
* The paper lacks a clear and actionable conclusion. While the findings are interesting, it remains unclear how these results can be directly applied to practical scenarios.

**Questions:**

The GGA algorithm shows promise, but it only works on the use of a limited number of adapters. This limitation is problematic given that adapters are usually lightweight and commonly stacked layer-wise in practical. As a result, the applicability of GGA in real-world scenarios is severely constrained.

---

> ### Author Response · Authors · 2024-11-20
> **Response to Reviewer BPzc**
>
> We appreciate the Reviewer's feedback on our work. We're glad that the Reviewer finds the topic of optimal adapter placement interesting and recognizes the potential of the GGA algorithm. We've addressed the specific concerns raised below:
>
> > The recurrent adapters require two forward processes, further deteriorating the efficiency of pre-trained models
>
> The motivation for introducing the recurrent adapters was to explore all potential pathways for information adaptation within a network. As we have discussed in Chapter 3 (line 269), the recurrent adapters introduce at most 50\% increase of the computational cost but do not introduce any additional parameters to the network. This makes them a compelling option for scenarios where an increase in computational time can be traded for improved accuracy, without requiring additional parameter resources. One can always opt to only use the diagonal and upper triangular parts of the connectivity matrix. To demonstrate this, we restricted the extended search space to not include any recurrent connections. Table 7 in the Appendix shows that the naive random search can still uncover placements that match the performance of Parallel Adapters using half of the  parameters without increasing inference cost.
>
> > Results in table 1 lack the comparison with related works.
>
> The aim of the experiment in Table 1 is to demonstrate that there exist adapter placements within the extended search space that can outperform the standard PA approach. This is why we compare our method with other adapter placement strategies. For a comparison of how adapters perform in relation to other PETL approaches such as LoRA or Prefix Tuning, please refer to the work that focused on that topic (e.g. [1,2]).
>
> > The paper lacks a clear and actionable conclusion. While the findings are interesting, it remains unclear how these results can be directly applied to practical scenarios.
>
> The key takeaways for practitioners from our paper are as follows:
>
> 1. Performing a random search within an extended adapter search space can effectively reduce the parameter count.
> 2. Since an adapter’s performance is often linked to the rank of its gradient, this metric can be leveraged to identify optimal adapter placements (as demonstrated by the GGA algorithm).
> 3. Adapter selection needs to be dataset-specific, as the best placements are unlikely to generalize across unrelated datasets.
>
> Crucially, our work establishes a strong connection between adapter placement and performance, paving the way for future research. This insight encourages further exploration into how this relationship can be harnessed to develop new guidelines for designing adapters.
>
> **Questions:**
>
> >The GGA algorithm shows promise, but it only works on the use of a limited number of adapters [...]
>
> By the use of GGA we can match or surpass the performance of the baseline approach with less adapters - consider, for instance Figure 6a, in which we outperform the PA approach with as few as 3 adapters, or Figure 6b in which we only need 12 adapters to match the performance of PA (that uses 24 adapter by default). In consequence, we are able to further reduce the parameter count. Note that even though adapters are inherently lightweight, further reducing their parameter count is far from pointless; it remains an active area of research with substantial ongoing work. Consider, for instance, scenarios where adapters need to be optimized and stored separately for a large number of disjoined tasks using the same pre-trained model. Additionally, reducing the number of parameters could facilitate fine-tuning on a single GPU, enabling a greater number of simultaneous runs or improving the accessibility of transfer learning to researchers and developers without access to extensive computational resources.
>
> We again thank the Reviewer for their valuable feedback and believe we have addressed their concerns. We remain available in case of any further questions  and kindly ask the Reviewer to consider our responses and reassess our work.
>
> *References*
>
> [1] He, Junxian, et al. "Towards a unified view of parameter-efficient transfer learning." arXiv preprint arXiv:2110.04366 (2021).
> [2] Zhang, Yuanhan, Kaiyang Zhou, and Ziwei Liu. "Neural prompt search." arXiv preprint arXiv:2206.04673 (2022).

---

> > ### Comment · Reviewer_BPzc · 2024-11-30
> > **While I appreciate the response, unresolved issues prevent me from raising my assessment**
> >
> > Thanks for your response to my review. While I appreciate the authors' efforts to address my concerns, my primary issues remain unresolved, particularly regarding the key takeaway, which diminishes my confidence in the work.
> >
> > Firstly, the authors have not adequately clarified the significance of reducing the parameter count of the adapter, given that adapters are already significantly smaller than the LLM itself. Whether the adapter has 1M or 10M parameters does not fundamentally impact its ability to be fine-tuned on a single GPU.
> >
> > Secondly, I previously highlighted the novelty of the GGA algorithm, but also noted its limited applicability. Unfortunately, the authors have not addressed this limitation in their response. While GGA is an interesting approach, its scope is constrained, and further improvements are needed.
> >
> > Finally, the dataset-specific nature of adapters is not a novel finding. While it may be challenging to find a universally applicable guideline, there should be some generalizable patterns that the authors could have explored.
> >
> > In conclusion, although the paper offers some insights, it remains incomplete in several areas. I cannot recommend the paper for acceptance in its current form and will therefore be lowering my score.

---

> ### Author Response · Authors · 2024-12-01
>
> The Reviewer raised three concerns in their response as a main basis for lowering their score. The first two address the significance and limitations of GGA, while the third pertains to the novelty of data-specific adapter placement. However these arguments are just not supported with evidence. We summarize why we believe reviewers arguments to be wrong below and hope the reviewer can provide more evidence if they disagree.
>
> - [Significance of reducing parameter count] - the Reviewer argues that adapters are already small, implying that hence there is no point in investigating methods that further increase their parameter efficiency. This is plainly not true as we explain in our response (paragraph “By the use of GGA [...]”) where we provided the reasoning on why reducing parameters in fine-tuning is a critical research focus. Furthermore, let us note that efforts to enhance parameter efficiency in transfer learning align with the broader goal of identifying the minimal parameters or resources needed to solve a given task. This paradigm lies at the core of the field of Parameter Efficient Transfer Learning, as evidenced by numerous studies exploring ways to improve the parameter efficiency of PETL methods (e.g., [1,2,3]). The Reviewer’s argument seems to undermine the entire premise behind this field.
>
> - [limitation of GGA] - In their review, the Reviewer writes that "the GGA algorithm shows promise, but it only works on the use of a limited number of adapters. This limitation is problematic given that adapters are usually lightweight and commonly stacked layer-wise in practical [...]” We responded by explaining that “by the use of GGA we can match or surpass the performance of the baseline approach with less adapters [...]” and provided arguments why this is important (see the point above).
>
> - [Dataset-specific nature of adapters is not a novel finding.] - The Reviewer claims that the data-specific aspect of adapter placement investigated in our work is not novel, yet offers no citations or references to back up this claim.
>
> In summary, the Reviewer asserts that the investigation into the optimal placement of adapters for a given task is not novel, yet provides no references to support this claim, while also dismissing the importance of further reducing the parameter costs of PETL methods. Unfortunately, we didn't hear  back from other reviewers or our AC to step-in to this discussion. We hope it sparks a constructive discussion among the reviewers and the AC, leading to a fairer evaluation of our work.
>
> References:
>
> [1] Karimi Mahabadi, Rabeeh, James Henderson, and Sebastian Ruder. "Compacter: Efficient low-rank hypercomplex adapter layers." Advances in Neural Information Processing Systems 34 (2021): 1022-1035.
> [2] Kopiczko, Dawid J., Tijmen Blankevoort, and Yuki M. Asano. "Vera: Vector-based random matrix adaptation." arXiv preprint arXiv:2310.11454 (2023).
> [3] Chen, Shoufa, et al. "Adaptformer: Adapting vision transformers for scalable visual recognition." Advances in Neural Information Processing Systems 35 (2022): 16664-16678.

---

### Official Review · Reviewer_6CsQ · 2024-11-03

**Soundness:** 3
**Presentation:** 3
**Contribution:** 3
**Rating:** 6
**Confidence:** 2

**Summary:**

This work investigates efficient strategies for placing adapters in transfer learning, specifically within the context of parameter-efficient transfer learning (PETL). A key finding is that the location of adapters within a network significantly impacts effectiveness, and the optimal placement depends on the specific task. By adapting pre-trained models to new tasks with minimal parameter adjustments, this study emphasizes maximizing effectiveness through strategic adapter placement rather than uniformly adding adapters across network layers.

**Strengths:**

1. The designs for the adapter in Equation 2 and the recurrent searching graph appear solid. It seems that using the recurrent approach is more effective than the parallel sequential or long-range approach.

2. Based on experiments, a single, well-placed adapter can significantly improve performance compared to the linear probe.

**Weaknesses:**

I think the paper could benefit if the author addresses the following question related to multiple adapters and the corresponding search and replacement algorithms:

1. How many adapters would achieve optimal performance? From Section 5.2, it appears that the authors aim to use multiple adapters with random search to improve transfer learning quality, yet it remains unclear how many adapters are necessary to achieve robust performance coverage. Based on Figure 6 and the GGA section, performance appears to continue increasing as more adapters are added. While I understand that the optimal number of adapters is task-dependent, is there a mathematical formalization that explicitly demonstrates how many adapters are sufficient? It seems that GGA is largely observational and based on gradient analysis. Is it possible that adding too many adapters could lead to an over-parameterized model, potentially causing a drop in accuracy?


2. The authors suggest that GGA is effective with a few adapters (as shown in Figure 6, where 12 GGA-based adapters achieve similar or higher performance). I believe it would be beneficial if the authors provided comparisons of computational complexity and memory usage for these methods. Specifically, what is the training complexity compared to the compressed approach in this paper? Calculating gradients may introduce additional time and memory costs, as it requires retaining the graph during training. Additionally, what is the inference complexity? For instance, to reach the same accuracy, full fine-tuning does not incur additional costs associated with adapters (which could treated as upper bound). How does GGA compare with other approaches in terms of these factors?

**Questions:**

Related to question 1 in terms of weaknesses, the authors suggest that the placement of adapters significantly impacts performance. In the case of multiple adapters, as shown in Figure 6, using the same number of adapters (e.g., 6, 12, 24) results in varied performance. The only difference is the placement of these adapters, while other conditions remain the same.

---

> ### Author Response · Authors · 2024-11-20
> **Response to Reviewer 6CsQ [Part 1/2]**
>
> We thank the Reviewer for the time and effort dedicated to evaluating our work. We are pleased to learn that the Reviewer considers our construction of the searching graph as solid, and has recognized the significance of the improved performance achieved through the strategic placement of adapters, as demonstrated in our experiments. We greatly appreciate this positive feedback. Below, we address the issues raised by the Reviewer in detail.
>
> > 1.How many adapters would achieve optimal performance?
>
> In Figures 6a and 6b we examined the effect of the number of adapters on the transfer performance. We observe that while adding more adapters initially enhances the results, the improvement eventually plateaus for GGA—the best-performing method—after 12 adapters. Due to this saturation we did not explore configurations with more than 24 adapters, which is the number used by the baseline Parallel Adapters (PA) approach. Moreover, adding more adapters often does not improve the test accuracy. For instance, on the SST2 dataset, using all $N^2=625$ adapters from the extended search space achieves a test accuracy of 93.81$\pm$0.76, which falls short of the 95.00$\pm$0.25 obtained by the best random selection of just 24 adapters (see the new Figure 15 in Appendix). This raises the question of whether peak performance can be achieved with fewer parameters. Our study demonstrates that by strategically selecting adapter locations, it is possible to reduce the number of adapters required.
>
> > is there a mathematical formalization that explicitly demonstrates how many adapters are sufficient?
>
> We agree that the topic of how many adapters are needed to achieve optimal performance is very interesting,  and in general touches on the broader problem of generalization capabilities in the context of transfer learning [1,2]. To the best of our knowledge, no existing research has theoretically explored the relationship between adapter capacity and its impact on transfer generalization. Even in the context of linear probing and full fine-tuning, establishing solid mathematical foundations remains an open area of inquiry [3,4,5]. Predicting the optimal number of adapters in advance is generally difficult and, as demonstrated by the experiments in Figures 6, may depend on various factors, such as the underlying task or the chosen search space.
>
> > Is it possible that adding too many adapters could lead to an over-parameterized model, potentially causing a drop in accuracy?
>
> As with any deep learning framework, introducing too many trainable parameters relative to the available training data can lead to issues with overparameterization. We do, in fact, observe one such situation in our study, as shown by the results for the Clevr-Count-1K dataset in Figures 1 and 3. Here, the full fine-tuning (FT) method, despite achieving the lowest training loss (below $10^{−4}$, see Table 6 in the Appendix), yields the worst test accuracy among all approaches. Except for this example, the datasets studied by us do not have this issue. Nevertheless, even in PETL, maintaining an appropriate balance between model complexity and the available data is an essential task of the researcher.
>
> > The authors suggest that GGA is effective with a few adapters (as shown in Figure 6, where 12 GGA-based adapters achieve similar or higher performance). I believe it would be beneficial if the authors provided comparisons of computational complexity and memory usage for these methods. Specifically, what is the training complexity compared to the compressed approach in this paper? Calculating gradients may introduce additional time and memory costs, as it requires retaining the graph during training.
>
> Please note that we compute the GGA scores (Equation 7 and 6) **only once**, after which GGA uses discounting to sample a set of adapters (see Algorithm 1 in the Appendix). Calculating the GGA scores for a network with $N^2$ possible placements involves computing the gradients from Equation 6 and evaluating Equation 7.  For the ViT-16/B model from the experiment in Figure 6, this cost amounts to approximately  $110.0$ TFLOPS (see the new Appendix J for details).  A forward pass of ViT-16/B model with batch size 128 is approximately 4.5 TFLOPS. Since we perform 20000 iterations during the adaptation (that amount to a total 90 PFLOPS, and this is considering only the forward passes, without the cost of backpropagation)  $F_{GGA}$ introduces only a negligible (0.12%) overhead in comparison.

---

> > ### Author Response · Authors · 2024-11-20
> > **Response to Reviewer 6CsQ [Part 2/2]**
> >
> > > Additionally, what is the inference complexity?
> >
> > If there are no recurrent adapters in the network, the inference complexity that comes with the use of the extended search space is the same as for the baseline Parallel Adapters approach (For a setup with batch size 128, rank 8 and 24 adapters this amounts to 15.4 GFLOPS (see Appendix J), which is negligible compared to the 4.5 TFLOPS required for the same batch size for a single forward pass of the ViT-16/B model).
> >
> > As discussed at the end of Chapter 3, the introduction of the recurrent adapters introduces an additional cost caused by our implementation of two passes through the network. The first pass is needed to propagate the $z_i$ up to the index $i$ which is the maximal value such that $(i,j)$ is in the adapter set and i>j (recall Equations 3-5).  If $i$ is the last layer, then the resulting cost is equal to the forward pass of a network with no adapters, and the total inference complexity increases by 50% compared to a setup with no recurrent adapters (see lines 269-278 and Appendix J).
> >
> > Recurrent adapters improve performance but come at the cost of increased computation. This trade-off between enhanced results and computational overhead makes them a useful, though optional, feature. As shown in Appendix Table 7, strategically positioning adapters within the connectivity matrix—without employing recurrent connections—can achieve performance comparable to Parallel Adapters while using only half the parameters and without increasing inference costs.
> >
> > Side note: We have realized that we had a typo in Equations 3-5: all $i$ should be replaced by $j$ and vice versa - we have updated the paper with the correct notation.
> >
> > We sincerely appreciate the Reviewer's feedback. We have diligently addressed all raised issues. Please let us know if you have any further questions. Additionally, we kindly ask the Reviewer to reconsider our work at their convenience.
> >
> > *References*
> >
> > [1] Blanchard, Gilles, et al. "Domain generalization by marginal transfer learning." Journal of machine learning research 22.2 (2021): 1-55.
> > [2] Lampinen, Andrew K., and Surya Ganguli. "An analytic theory of generalization dynamics and transfer learning in deep linear networks." arXiv preprint arXiv:1809.10374 (2018).
> > [3] Tripuraneni, Nilesh, Michael Jordan, and Chi Jin. "On the theory of transfer learning: The importance of task diversity." Advances in neural information processing systems 33 (2020): 7852-7862.
> > [4] Williams, Jake, et al. "Limits of transfer learning." Machine Learning, Optimization, and Data Science: 6th International Conference, LOD 2020, Siena, Italy, July 19–23, 2020, Revised Selected Papers, Part II 6. Springer International Publishing, 2020.
> > [5] Hanneke, Steve, and Samory Kpotufe. "A More Unified Theory of Transfer Learning." arXiv preprint arXiv:2408.16189 (2024).

---

### Author Response · Authors · 2024-11-25

We sincerely thank all the Reviewers once again for their valuable feedback. We have worked to address all the concerns raised and have updated our paper accordingly. As the discussion period comes to an end, we kindly ask the Reviewers to reevaluate our work in light of our responses. We remain available to address any further questions or considerations.

---

### Meta-Review · Area_Chair_HZS7 · 2024-12-20

**Metareview:**

The paper proposes a method to place adapters based on the tasks with the premise that adaptation of the network is task dependent. This is done through a method called Gradient Guided Adapters. As pointed out by one of the reviewers,  It is true that there is a rich literature of task specific adaptation for instance (a very small selection)
Budget-Aware Adapters for Multi-Domain Learning
Rodrigo Berriel, Stéphane Lathuilière, Moin Nabi, Tassilo Klein, Thiago Oliveira-Santos, Nicu Sebe, Elisa Ricci

TAIL: Task-specific Adapters for Imitation Learning with Large Pretrained Models
Zuxin Liu, Jesse Zhang, Kavosh Asadi, Yao Liu, Ding Zhao, Shoham Sabach, Rasool Fakoor

and many more. Hence, the paper does seem to have a gap in the prior work and comparison in that area. However, the uses of recurrent and adapters is a novelty of the method. Hence it is unclear what the efficacy of the proposed method is with respect to the existing methods. Further, there are some other benchmarks on which these can be compared against for instance the ImageNet to Sketch benchmark and the domainnet benchmark. While these are older, they will help compare across many methods that have used them.

Given this fact as well as the reviewer ratings, this paper unfortunately is not ready for the ICLR audience.

**Additional Comments On Reviewer Discussion:**

The rebuttal was not very robust. However, reviewer BPzc did respond to the comments. I think the biggest negative for the paper was the lack of comparisons and literature survey to existing work.

---

### Decision · Program_Chairs · 2025-01-22

Reject